Shifts in stability and control effectiveness during evolution of Paraves support aerial maneuvering hypotheses for flight origins

Evangelista Dennis 1 5 devangel77b@gmail.com
Cam Sharlene 1
Huynh Tony 1
Kwong Austin 2
Mehrabani Homayun 2
Tse Kyle 3
Dudley Robert 1 4
1 Department of Integrative Biology, University of California , Berkeley, CA , USA
2 Department of Bioengineering, University of California , Berkeley, CA , USA
3 Department of Mechanical Engineering, University of California , Berkeley, CA , USA
4 Smithsonian Tropical Research Institute , Balboa , Panama
Hutchinson John
5 Current affiliation: University of North Carolina at Chapel Hill, NC, USA

Electronic publication date: 2014 Oct 16
Publication date: 2014
Volume: 2
Electronic Location ID: e632
Received 2014 Jul 10; Accepted 2014 Sep 30
Copyright: © 2014 Evangelista et al.
Copyright year: 2014
Copyright holder: Evangelista et al.
License: This is an open access article distributed under the terms of the Creative Commons Attribution License, which permits unrestricted use, distribution, reproduction and adaptation in any medium and for any purpose provided that it is properly attributed. For attribution, the original author(s), title, publication source (PeerJ) and either DOI or URL of the article must be cited.
License URL: https://creativecommons.org/licenses/by/4.0/

Keywords: Stability, Control effectiveness, Maneuvering, Flight, Evolution, Paraves, Biomechanics, Directed aerial descent

Funding: NSF DGE-0903711 University of California Museum of Palaeontology (UCMP) DE was supported by an NSF Minority Graduate Research Fellowship, UC Chancellor’s Fellowship, and NSF Integrative Graduate Education and Research Traineeship (IGERT) #DGE-0903711. TH was supported by the University of California Museum of Palaeontology (UCMP). The funders had no role in study design, data collection and analysis, decision to publish, or preparation of the manuscript.

==============================
The capacity for aerial maneuvering was likely a major influence on the evolution of flying animals. Here we evaluate consequences of paravian morphology for aerial performance by quantifying static stability and control effectiveness of physical models for numerous taxa sampled from within the lineage leading to birds (Paraves). Results of aerodynamic testing are mapped phylogenetically to examine how maneuvering characteristics correspond to tail shortening, forewing elaboration, and other morphological features. In the evolution of Paraves we observe shifts from static stability to inherently unstable aerial planforms; control effectiveness also migrated from tails to the forewings. These shifts suggest that a some degree of aerodynamic control and capacity for maneuvering preceded the evolution of a strong power stroke. The timing of shifts also suggests features normally considered in light of development of a power stroke may play important roles in control.

Introduction

Regardless of how aerial behavior originates, once airborne an organism must control (Smith, 1952) its orientation and position in order to safely navigate the vertical environment (e.g., directed aerial descent, Dudley & Yanoviak, 2011). Such abilities are present even in taxa with no obvious morphological adaptation for flight (such as Cephalotes ants, Munk, 2011; geckoes, Jusufi et al., 2008; Jusufi et al., 2011; stick insects, Zeng, 2013; and human skydivers, Cardona et al., 2011; Evangelista et al., 2012). At low speeds, such as at the start of a fall or jump, inertial mechanisms (Jusufi et al., 2008; Jusufi et al., 2011) allow for rolling, pitching, and yawing. As speeds increase (or as appendages grow in area), aerodynamic mechanisms of control can be employed. Body and appendage configuration and position affect both stability, the tendency to resist perturbations, as well as production of torques and forces for maneuvering (control effectiveness). In the four-winged Early Cretaceous Microraptor gui, changes in planform, such as alternative reconstruction postures or removal of leg and tail feathers, alter stability and the control effectiveness of appendages (Evangelista et al., 2014b). Furthermore, appendage function can shift entirely according to the aerial environment (e.g., asymmetric wing pronation producing yaw at high glide angle versus roll at low glide angle) or even completely reverse function (Evangelista et al., 2014b). Such aerodynamic results are exciting but are based on a single specimen (Xu et al., 2003). Stronger conclusions can be drawn from comparative study of several forms within a phylogenetic context.

One obvious trend in avian evolution is the transition from long tails and feathered legs in early forms (Xu et al., 2003; Hu et al., 2009; Longrich, 2006; Christiansen & Bonde, 2004; Zheng et al., 2013; O’Connor et al., 2013; Pittman et al., 2013) to later (including extant) forms for which the skeletal tail has fused into a short pygostyle and both asymmetric and symmetric flight feathers are absent from the legs. Functional consequences of this shift for aerial maneuvering remain speculative (Smith, 1953; Beebe, 1915; Thomas, 1997). Similarly, changes in the pectoral girdle have been assumed to enhance a powered downstroke (Gauthier & Padian, 1985; Benton, 2005), but may also have influenced maneuvering by shifting the center of body mass (Allen et al., 2013) or in enabling the production of wing asymmetries. With the exception of Huynh et al. (2011), previous aerodynamic studies tend to focus on lift and drag coefficients and glide angles and specific postures (Chatterjee & Templin, 2007; Alexander et al., 2010; Koehl, Evangelista & Yang, 2011; Dyke et al., 2013), with maneuvering only considered rarely and in single taxa (Longrich, 2006; Hall et al., 2012; Evangelista et al., 2014b).

To examine these patterns and to test adaptive hypotheses (Padian, 2001), we can use model tests to quantify the effect of shape on static stability and control effectiveness (Evangelista et al., 2014b; Koehl, Evangelista & Yang, 2011; McCay, 2001), using specimens sampled from paravian (Xu et al., 2011) and early avialan (Gauthier & Padian, 1985) evolution. We focus specifically on static stability and control effectiveness; while lift and drag are expected to be important in flight evolution, they have been addressed adequately in previous literature (Evangelista et al., 2014b; Dyke et al., 2013; Koehl, Evangelista & Yang, 2011; Alexander et al., 2010). While the capacity to generate aerodynamic forces for weight support (Dial, 2003; Dial, Jackson & Segre, 2008; Heers, Dial & Tobalske, 2014) was almost certainly evolving, we consider here the ability to control aerial behavior (Dudley & Yanoviak, 2011). The presence or absence of stability in the various axes and the control effectiveness of the appendages should correspond to changes in major morphological features (shortening of the tail, enlargement of the forewings) to enhance aerodynamic performance, however, previous work has not yet identified the patterns. We hypothesize that stability and control are present early in the evolution of flight (Dudley, 2002; Dudley & Yanoviak, 2011); this can be tested by examining the patterns of stability and control. Morphologically, we would predict organisms would have some amount of surface area (Xu et al., 2003; Hone et al., 2010; Xu et al., 2011; Zheng et al., 2013; Zhang & Zhou, 2004); though it may not be large, it should provide some amount of stability. At an early stage, organisms may possess only a weak power stroke. With some appendages (particularly, forelimb wings), they may initially have limited ability to generate aerodynamic forces due to reduced speed or feather porosity and flexibility (Nudds & Dyke, 2009; Nudds & Dyke, 2010; Heers, Tobalske & Dial, 2011). We would also predict additional appendages (tails or legs) with sufficient flexibility (Kambic, Roberts & Gatesy, 2014) and inertia (Jusufi et al., 2008; Jusufi et al., 2011), in combination with aerodynamic forces, to generate moments and effect maneuvers (Dudley, 2002; Dudley & Yanoviak, 2011). These are all present at the base of the tree of the taxa tested. Furthermore, as the clade evolves, the absence of stability, coupled with the presence of large control effectiveness, could be used to infer the presence of strong closed-loop neuromuscular control. The absence of control effectiveness would suggest a lack of control, as even with feedback an ineffective surface cannot generate the necessary forces and torques.

Alternatively, both stability and control may have been absent early in the evolution of flight, only appearing after a strong and bilaterally symmetric power stroke evolved; or very stable platforms with a lack of effective control mechanisms may be observed. For these alternatives, morphologically we might expect skeletal features with large surfaces and strong muscle attachments including an early carinate sternum to provide the power stroke, and wrist joints and shoulders with highly restricted ranges of motion oriented to provide a fundamental flight stroke oriented to gravity with limited degrees of freedom. For the case of stability without control, we may also expect rigidly fused/fixed stabilizers located posteriorly, and without mobility at the base. Other then a semilunate carpal assumed to enable a flight stroke as an exaptation of a raptorial strike mechanism (Padian, 2001), the morphological features we predict under a power-stroke first/stability and control later hypothesis are not observed in the taxa tested here (Benton, 2005; Gatesy & Baier, 2005; Zhou & Li, 2010; Turner, Mackovicky & Norell, 2012; O’Connor et al., 2012; O’Connor et al., 2013). Our hypothesis could still be falsified via aerodynamic testing that shows either a lack of stability and control or a lack of control; it is possible for some appendage motions or positions to ineffective or unstable, as in the case of certain sprawled postures and leg movements in Microraptor (Evangelista et al., 2014b, although these were also anatomically infeasible).

Smith (1952), in outlining the potential importance of static stability and control effectiveness, called for measurements to test his assertions: “if the conclusions of this paper are accepted the study of the remains of primitive flying animals, and even experimental studies on full-scale models of them, will acquire a special importance as throwing new light on the functional evolution of nervous systems (Smith, 1952)”. Subsequent studies of stability have been limited to computational studies with regard to aerodynamic function, etc. (Gatesy & Dial, 1996; Taylor & Thomas, 2002; Thomas & Taylor, 2001). Computational estimates are useful, but when checked against models (Evans, 2003) or live animals (Clark, 2010) there are occasionally unexpected deviations from aerodynamic theory developed for small angles of attack and airplane-like morphologies. Therefore, we measured static stability and control by measuring the aerodynamic moments exerted on physical models of different specimens in a wind tunnel, including at large angles of attack.

Materials and Methods

Model construction

We constructed models (8 cm snout-vent length, Figs. 1–2) of four extant birds and seven fossil paravians (Xu et al., 2011), encompassing five avialans (Gauthier & Padian, 1985), Microraptor (Xu et al., 2003) and Anchiornis (Hu et al., 2009) (Fig. S1), using 3D printing. Fossils were selected to sample phylogenies available in 2011 (when the work was done), although an eighth paravian, Zhongornis (Gao et al., 2008), was later dropped due to questions about its phylogenetic position and because the specimen was identified to be a juvenile. Additional information regarding the morphology of Microraptor, Jeholornis, and Sapeornis has become available since the measurements were done; the effect of the new information is addressed in the discussion. To explore parallel evolution and for calibration, we also constructed models of three pterosaurs, two bats (Bitbucket), and two artificial test objects (sphere and weather vane) (Fig. 2). Construction methods closely followed those of Koehl, Evangelista & Yang (2011), Evangelista et al. (2014b), Evangelista (2013), Munk (2011) and Zeng (2013). Solid models were developed in Blender (The Blender Foundation, Amsterdam), closely referencing published photographs of fossils and reconstructions from the peer-reviewed literature and casts of Archaeopteryx to match long bone, axial skeleton, and body proportions. Modeling was also guided by Starling (Sturnus vulgaris) dissections, preserved specimens, and vertebrate paleontology and anatomy/functional morphology texts (Benton, 2005; Liem et al., 2000). To create the models, photos of the fossils were imported into Blender and used to guide digital sculpting of a series of 3-D meshes representing the head, torso, tail and limbs of each taxon. Each mesh was constrained to maintain left–right symmetry and stretched in the antero-posterior, dorsoventral, and lateral axes to approximately match the proportions of the fossil or reconstruction. Specific target points used included the length of all limb elements (humerus, radius/ulna, and manus; femur, tibiotarsus, tarsometatarsus, and pes), length of the skull, and distances along the axial skeleton (occipital to pectoral; pectoral to pelvic, and tail) as well as an approximate depth of the ribcage where available. In some cases, some of these targets were estimated. Meshes were provided with holes to accept a 26 gauge armature and mounts for the force sensor (discussed below). This process was used to provide more repeatable and replicable results than hand sculpting by eye with polymer clay (as used in Koehl, Evangelista & Yang, 2011; Evangelista et al., 2014b). Specific species and the corresponding specimen and references include Anchiornis (LPM B00169, Hu et al., 2009), Microraptor (IVPP V13352, Xu et al., 2003), Archaeopteryx (Berlin specimen as reconstructed in Longrich, 2006), Jeholornis (IVPP V13274 and 13553, Zhou, Zhang & Science, 2002; Zhou & Zhang, 2003b), Sapeornis (IVPP V13275, Zhou & Zhang, 2003a), Zhongjianornis (IVPP V15900, Zhou & Li, 2010), and Confuciusornis (Hou et al., 1995; Chiappe et al., 1999; Chiappe et al., 2008). The .STL files used to create the models are available for download to researchers wishing to replicate our models.

Figure 1 Model construction.

Models were developed in Blender (A) from fossils (Archaeopteryx shown) and constructed using previous methods (McCay, 2001; Koehl, Evangelista & Yang, 2011; Munk, 2011; Evangelista et al., 2014b). Models for fossil paravians studied are shown in (B)–(H), scale bars indicate 8 cm snout-vent length. Anchiornis (B) (LPM B00169, Hu et al., 2009), hind limbs rotated out of test position to show plumage for illustration only. Microraptor (C) (IVPP V13352, Xu et al., 2003; tape covering proximal wing to body not shown). Archaeopteryx (D) (Berlin specimen as reconstructed in Longrich, 2006). Jeholornis (E) (IVPP V13274, 13553, Zhou, Zhang & Science, 2002; Zhou & Zhang, 2003b). Sapeornis (F) (IVPP V13275, Zhou & Zhang, 2003a). Zhongjianornis (G), (IVPP V15900, Zhou & Li, 2010). Confuciusornis (H) (Hou et al., 1995; Chiappe et al., 1999; Chiappe et al., 2008).

Figure 2 Testing and measurement of moments.

Models were tested (A) using previous methods (McCay, 2001; Koehl, Evangelista & Yang, 2011; Evangelista et al., 2014b). For simple cases such as a sphere or a weather vane, the relationship between slope and stability (B) is observed by plotting pitching moments versus angle of attack; negative slopes indicate restoring moments and stability while positive slopes indicate instability. Moments for sphere are not statistically different than zero, indicating marginal stability as expected, further validating the methods.

Models were printed using a 3D printer (ProJet HD3000; 3D Systems, Rock Hill, SC), then mounted on 26 gauge steel armatures with felt or polymer clay filling in gaps between printed parts where flexibility was needed for repositioning. Wings were constructed using methods described in Koehl, Evangelista & Yang (2011) and Evangelista et al. (2014b). Wings were traced from the published reconstructions in peer-reviewed publications (Hu et al., 2009; Longrich, 2006; Hou et al., 1995; Chiappe et al., 1999; Zhou, Zhang & Science, 2002; Zhou & Zhang, 2003b; Xu et al., 2003; Zhou & Zhang, 2003a; Zhou & Li, 2010), although this procedure was subject to uncertainty in preservation and reconstruction inherent even in work by acknowledged paleontological experts in peer-reviewed publications. Wings were printed on paper and cut. Monofilament stiffening was added along feather rachises and the wings were attached to 26 gauge steel limb and tail armatures (scaled as described above) using surgical tape (3M, St. Paul, MN). Surgical tape was also used where necessary to secure the proximal wing to the body and to repair light damage during testing. This procedure was found in previous work to match more laborious and less repeatable application of manually attached real bird feathers (Koehl, Evangelista & Yang, 2011) for fixed-wing tests.

Models produced in this manner provide an approximation to the general planform but do not replicate specific skeletal elements (via X-ray scanning and 3D printing) as has since become possible; neither the methods nor the fossils were available to us when the work was done. While they may not perfectly mimic real animals, they do allow examination of how shape affects stability and control. Perfect mimicry is not possible for extinct organisms reconstructed from single or a few fossils; such a limitation is inherent in all modeling studies. We suggest that, by using multiple models in a phylogenetic context, we provide more robustness than does the case of a single model (as in all prior work). Also, the use of such techniques is justified based on benchmarking against extant animals and micro air vehicles at high angles of attack (discussed below).

Body posture and appendage position

Fossil paravian models were positioned with wings spread and legs extended back (Xu et al., 2004; Evangelista et al., 2014b) (Figs. 1–3). The aim of this study was to examine maneuvering within several species rather than the posture of one; accordingly we used a legs-back posture seen in extant birds and also reasonably well supported for fossils (Xu et al., 2004; Davis, 2008), with leg feathers where present. While alternative postures have been considered (specifically in Microraptor, as reviewed in Koehl, Evangelista & Yang, 2011; Evangelista et al., 2014b), some may be infeasible, notably sprawled (Xu et al., 2003); others are not observed in extant birds, such as biplane configurations (Chatterjee & Templin, 2007; Alexander et al., 2010) or less extreme sprawled leg positions (Dyke et al., 2013); while legs-down positions (Huynh et al., 2011; Habib et al., 2012; Hall et al., 2012; Dyke et al., 2013; Evangelista et al., 2014b) appear to have very high wing loading compared to alternatives like legs-back.

For control effectiveness, we tested fixed static appendage movements previously identified as being aerodynamically effective (Evangelista et al., 2014b; Evangelista, 2013): asymmetric wing pronation and supination, wing tucking, symmetric wing protraction and retraction, and dorsoventral and lateral movements of the tail (Fig. 3). The angular extent of each movement tested is shown on Fig. 3. To avoid confusion, we present data for specimens as described with all appendages present; artificial manipulations (such as removal of tail and leg surfaces) were discussed in Evangelista et al. (2014b) and Evangelista (2013).

Figure 3 Appendage movements tested to determine control effectiveness.

Light gray indicates baseline posture, dark gray indicates appendage deflection. Appendage movements were selected based on those observed to be effective in previous work (Evangelista et al., 2014b), including (A) symmetric wing protraction (e.g., wing sweep to ±45°); (B) tail dorsiflexion to ±15°; (C) tucking of one wing; (D) tail lateral flexion to 30°; and (E) asymmetric wing pronation/supination to 15° (e.g., left wing pitched down, right wing pitched up).

Models were mounted at the estimated center of mass (COM) for the baseline body posture. The estimate was formed in Blender assuming a uniform density for the posed model, as in Allen et al. (2013). While we did not duplicate the same sensitivity analyses as Allen et al. (2013), we recognize that the COM estimate could vary up to 3–5% of the body length, or by smaller amounts for the variable load associated with appendage movements; this uncertainty is usually within the bounds of coefficient estimates identified as marginally stable. All scale models were tested on a sting, zeroed before each measurement, and thus the mass of the model does not affect testing. Mass and wing loading of the organism, while important for estimating speed from lift and drag coefficients, do not directly affect the nondimensional coefficients presented here.

Wind tunnel testing

Wind tunnel testing used previous methods (Evangelista et al., 2014b), with a six-axis sensor (Nano17; ATI, Apex, NC) mounted to a 0.5 inch (12.7 mm ) damped sting exiting the model downwind at the center of mass (Fig. 2). In some measurements, a 2 mm steel extension rod or a 3 mm acrylic plate were used to avoid geometric interferences and to keep the sting several diameters away and downstream of aerodynamic surfaces. The sensor was zeroed immediately before each measurement, eliminating model deadweight effects. Models were tested in an open-circuit Eiffel-type wind tunnel with an 18 × 18 × 36 inch (45.7 × 45.7 × 91.4 cm) working section (Engineering Laboratory Design, Lake City, MN). Testing at 6 m s−1 resulted in a Reynolds number of ∼32,000 for all models, matching full scale for Anchiornis, Archaeopteryx, Sapeornis, Zhongjianornis, and Confuciusornis.

Under the test conditions, the aerodynamic coefficients of interest are reasonably constant with Reynolds number, Re = UL/ν, where L here is the snout-vent length and ν is the kinematic viscosity of air (Evangelista et al., 2014b; Evangelista, 2013). Early in the evolution of animal flight, organisms likely flew at moderate speeds and high angles of attack (Evangelista et al., 2014b; Dyke et al., 2013) where flows appear like bluff body turbulent flows (in which coefficients are largely independent of Re, for 103 < Re < 106). In previous work (Koehl, Evangelista & Yang, 2011; Evangelista et al., 2014b), we performed a sweep of wind tunnel speed, to examine Re from 30,000 to 70,000, to validate that scale effects were not present.

As additional support for this approach, tests for maneuvering bodies are nearly always tested at well below full scale Re, e.g. the largest US Navy freely-maneuvering model tests are well below 13-scale. Our methods were also previously benchmarked using model tests at full scale Re of gliding frogs (Emerson, Travis & Koehl, 1990; McCay, 2001) (repeated for comparison), Draco lizards, Anna’s Hummingbirds (Calypte anna) in glide and extreme dive pullout maneuvers, hummingbird body shapes in hovering (Sapir & Dudley, 2012), and reduced-scale tests of human skydivers compared to actual data (Cardona et al., 2011; Evangelista et al., 2012); while at Re ∼ 1000, our modeling methods have been benchmarked against extant winged seeds. Perching aerial robots, developed to test control algorithms, have shown good agreement between fully 3D robots and flat plate models with the same planform (Roberts, Cory & Tedrake, 2009; Hoburg & Tedrake, 2009; Tangler & Kucurek, 2005). Results (Evangelista et al., 2014b) for lift and drag coefficients using our method agreed with those for full-scale Microraptor models in the other modeling tests (Dyke et al., 2013; Alexander et al., 2010); our Jeholornis model was at lower Re than Microraptor and is of similar shape.

Sensor readings were recorded at 1000 Hz using a data acquisition card (National Instruments, Austin, TX) (Evangelista et al., 2014b). The sting was mounted to a servo (Hitec USA, Poway, CA) interfaced to a data acquisition computer, using an Arduino microcontroller (SparkFun, Boulder, CO) and specially written code in Python and R (R Core Team, 2014), to automate positioning and measurement of windspeed and whole-body force/torque. Raw measurements were rotated to a frame aligned with the wind tunnel and flow using the combined roll, pitch, and yaw angles by multiplication with three Euler rotation matrices; translation from the sensor to the model COM was also included. Transformed measurements were averaged over a one-minute recording. We then computed non-dimensional force and moment coefficients, static stability coefficients, and control effectiveness (McCay, 2001; Evangelista et al., 2014b; McCormick, 1995). Three series, varying pitch, roll, and yaw, were conducted at 5° increments. Using the automatic sting, we obtained 13,792 measurements, with at least five replicates for 18 models in 247 total positions: generally 5 each in pitch (88 total), 2 each in roll for two angles of attack (69 total), and 3 each in yaw for two angles of attack (92 total). Test positions are indicated in Fig. 3.

Static stability was measured by examining the sign of the slope ∂Cm/∂α (positive slope is unstable, negative stable, zero marginally stable, see Figs. 2B and 5) of the non-dimensional pitching moment coefficient Cm near fixed points as the body was subjected to small deflections dα (Evangelista et al., 2014b; McCay, 2001; McCormick, 1995): (1) pitching moment M=0.5ρU2CmλS

where U is tunnel speed, λ is the snout-vent length, and S is planform area. S accounts for the overall area of the shape (as in lift and drag coefficients), while λ is a length scale needed dimensionally to obtain moments; use of snout-vent length is consistent with other previous work (McCay, 2001; Koehl, Evangelista & Yang, 2011; Evangelista et al., 2014b). Control effectiveness (∂Cm/∂δ, (Etkin & Reid, 1996; McCay, 2001; Evangelista et al., 2014b)) was measured by deflecting appendages (Fig. 3) by an amount dδ and examining the change in pitching moment coefficient. Both are unitless (rad−1). Roll and yaw were similarly calculated from series of varying roll angles or headings, respectively, with the static stability and control effectiveness partial derivatives taken for the roll (0.5ρU2CrλS) and yaw (0.5ρU2CyλS) moment coefficients. Stability (eight quantities) was computed for three axes (pitch, roll, and yaw) at low (15°) and high (75°) angle of attack as well as (for pitch only) at 0° and at pitching equilibrium. Control effectiveness (12 quantities) was similarly computed for three axes and at low and high angle of attack for the movements depicted in Fig. 3.

A first-order estimate of maneuvering is obtained from considering the two together and a biomechanical trade-off is apparent: a stable object can resist perturbations from the environment with minimal control effort but will also have difficulty in changing direction (which might be necessary to accomplish aerial righting, navigate in cluttered forests, seek resources or avoid predators) (Smith, 1952; Dudley & Yanoviak, 2011; Taylor & Thomas, 2002). The metrics underestimate maneuvering in very dynamic cases (high advance ratio flapping or where second-order damping terms become important; Sachs, 2005), but are adequate for quasi-static maneuvers. Locomotion is a complex task, and passive stability is often exploited where possible to reduce control effort (Jindrich & Full, 2002; Kubow & Full, 1999; Ting, Blickhan & Full, 1994); conversely, passive instability may be exploited in extreme (and likely elective) maneuvers. The absence of stability, coupled with the presence of large control effectiveness, would suggest the presence of strong closed-loop neuromuscular control. The absence of control effectiveness would suggest a lack of control, as even with feedback an ineffective surface cannot generate the necessary forces and torques. Thus, while the full control abilities of an extinct form are difficult if not impossible to fully enumerate, the simple metrics here provide a useful proxy.

Phylogenetic comparisons

We used the phylogeny shown in Fig. 4A. A Nexus file without branch lengths (bitbucket.org/devangel77b/comparative-peerj-supplemental) was assembled from published phylogenies of the study taxa. For paravians, the strict consensus of Zhou & Li (2010), Li et al. (2010) and O’Connor, Chiappe & Bell (2011) was used, with the family relationships of Cracraft et al. (2004) used to fill in the extant birds. Multiple sources were used because of differences in single species being included, but otherwise sources showed the same topology (reviewed in Turner, Mackovicky & Norell, 2012). While further revisions to the phylogenetic relationships have been discussed (depicted in Fig. 4B), (see Xu et al., 2011; Godefroit et al., 2013; Turner, Mackovicky & Norell, 2012; O’Connor et al., 2013), they do not appear to alter the patterns in stability and control effectiveness; trees from Xu et al. (2011), Godefroit et al. (2013), Turner, Mackovicky & Norell (2012) and O’Connor et al. (2013) are in the .nex file available at bitbucket.org/devangel77b/comparative-peerj-supplemental. Mapping, as outlined in Padian (2001), of discrete maneuvering traits was performed in Mesquite (Maddison & Maddison, 2010) with the built-in ancestral state reconstruction routines using unordered parsimony. The aerodynamic measurements were coded into a matrix giving the aforementioned eight discretized stability values and 12 discretized control effectiveness values. Stability values were coded as stable (slope < 0), marginal (slope = 0), or unstable (slope > 0) based on whether the 75% confidence interval of ∂C/∂α measurements included zero or not. The discretized control effectiveness values were obtained from the measurements by thresholding based on the moment necessary to overcome measured weather vane stability (dC/dδ > 0.09 was coded as effective; <0.09 coded as ineffective), or equivalently, to cause a displacement of the center of aerodynamic pressure of about 10% of total length.

Figure 4 Phylogenies.

(A) Phylogeny used in analyses, assembled from strict consensus of Zhou & Li (2010), Li et al. (2010), O’Connor, Chiappe & Bell (2011) for paravians and family relationships in (Cracraft et al., 2004) for extant birds. (B) Updated phylogeny from Turner, Mackovicky & Norell (2012) with revised position of Sapeornis, changes shown in blue. Additional proposed phylogenies (Godefroit et al., 2013; Xu et al., 2011), which alter the position of Archaeopteryx, are available in the .nex file at bitbucket.org/devangel77b/comparative-peerj-supplemental. Nodes 1–4 are discussed further in the text and are provided for reference between the trees.

Results

Representative aerodynamic measurements for pitching stability and control effectiveness are given in Fig. 5 for six paravians and two pterosaurs. As discussed below, Fig. 5 illustrates similarity in pitching moments, stability, and control effectiveness for long- (monotonically decreasing lines in Fig. 5A) versus short-tailed (humped curves in Fig. 5B) forms. In Fig. 5, pitching moment coefficient is plotted as a function of angle of attack for three conditions: tail at 0° (middle blue), tail up 15° (light blue), and tail down 15° (dark blue). Spread between these lines (yellow shaded box in Fig. 5A) indicates the presence of control effectiveness. The slope of the lines indicates stability, here marked at equilibrium Cm = 0. Yellow negative slope is stable, red positive slope is unstable; a zero slope would be marginal stability. Control effectiveness and stability determined in this manner were recorded for all test conditions and taxa; tables of all aerodynamic measurements are provided in Supplemental Information 1. All aerodynamic measurements were coded into both discretized and continuous character matrices (bitbucket.org/devangel77b/comparative-peerj-supplemental), which were then mapped onto a phylogeny (assembled from Zhou & Li, 2010; Li et al., 2010; O’Connor, Chiappe & Bell, 2011; Cracraft et al., 2004) to examine the evolution of static stability and control effectiveness.

The discretized character states for pitch, roll, and yaw are shown in Figs. 6–8. All trees show the most parsimonious reconstruction. Trees for each individual character are given, followed by a summary tree with all characters. For all trees, the alpha transparency (how faded or solid the line appears) indicates character state, while for the summary tree, color indicates characters. The trees are shown to explicitly illustrate the patterns of aerial maneuvering for several different character states within a phylogeny (Padian, 2001). As discussed below, the results are most consistent with presence of maneuverability early in evolution; stability and control effectiveness also reflect changes in morphology or glide angles at which control is possible. Alternate phylogenies (Xu et al., 2011; Godefroit et al., 2013; Turner, Mackovicky & Norell, 2012; O’Connor et al., 2013) are in the .nex file at bitbucket.org/devangel77b/comparative-peerj-supplemental, but do not alter the patterns seen. Each trait is subject to uncertainties of measurement (Supplemental Information 1), equipment failure, the limitations of ancestral state reconstructions with unordered parsimony, and uncertainty in the phylogenies, however, in the aggregate the results show consilience (later taxa lines are solid) among pitch and yaw traits as discussed further below.

Figure 5 Representative aerodynamic measurements for pitching stability and control effectiveness.

All plots show nondimensional pitching moment coefficient as a function of angle of attack. Long-tailed taxa (A) have a stable equilibrium point around 10–25° (yellow line) and the tail is effective in generating pitching moments at low angles of attack (pale yellow box indicates measurable moments for given tail deflections). In short-tailed taxa (B), including extant Larus, the equilibrium point at 0–5° is unstable (red line) and the tail control effectiveness is reduced (no measurable moments for the given tail deflections). Examples drawn from pterosaurs (Rhamphorhynchus and Pteranodon) illustrate similar patterns in phylogenetically distant taxa with contrasting tail lengths.

Figure 6 Evolution of pitch stability and control effectiveness.

Trees show (A) stability at equilibrium; (B) control effectiveness of the tail in pitch; (C) control effectiveness of symmetric wing protraction/retraction. Ancestrally, taxa are stable in pitch (A) and possess large, highly effective tails (B) but only moderately effective wings (C). Stability and tail control effectiveness are lost as tails shorten (AB, node 1), but more effective wings (C, node 1) are able to provide control. Control migrates from the reduced tail to the wings, which become larger and are associated with skeletal features that would enhance control and the production of left–right and fore-aft asymmetries.

Figure 7 Evolution of roll stability and control effectiveness.

Trees show (A) roll stability at low angle of attack; (B) roll stability at high angle of attack; (C) control effectiveness of asymmetric wing tucking in roll; (D) all of these together. Taxa are stable at high angle of attack (B), but mostly unstable at low angle of attack due to symmetry (A; Sapeornis and Confuciusornis marginal). Asymmetric wing tucking is always effective in roll (C). Thus, as animals developed the ability to fly at reduced body angles of attack, perhaps in shifting from steep-angle directed aerial descent (B) to shallower angles (A), more active control of roll would have been necessary. Ancestrally, inertial modes of the tail (Jusufi et al., 2008; Jusufi et al., 2011) would also have been available to assist the forewings, with function taken on solely by the forewings as tail inertia is reduced in derived taxa (after node 2).

Figure 8 Evolution of yaw stability and control effectiveness at high angle of attack (A)–(C) and at low angle of attack (D)–(G).

Trees show (A) yaw stability at high angle of attack; (B) tail and asymmetric wing pronation/supination control effectiveness; (C) yaw characters at high angle of attack together; (D) yaw stability at low angle of attack; (E) tail control effectiveness; (F) asymmetric wing control effectiveness; and (G) yaw characters at low angle of attack together. At high angles of attack (A)–(C), taxa are mostly marginally stable as might be expected for high angles (e.g., at 90° angle of attack all forms are marginal). Asymmetric pronation/supination of the wings are always effective in generating yaw at high angles of attack. At low angles of attack (D)–(G), by contrast, long-tailed taxa are stable and can control yaw with the tail. As tails reduce in size (nodes 1–2), taxa become unstable in yaw at low angles of attack and lose the ability to control yaw with the tail as well as any assistance from inertial modes of the tail. However, asymmetric movements of the wings are effective in producing yaw throughout the evolution of this clade, and control would thus have shifted from the tail to the forewings, paralleling the shifts seen in pitch.

Discussion

Additional information and new reconstructions

The aerodynamic tests described here were performed in the fall of 2011; since then, more information has become available leading to more detailed reconstructions and phylogenies being proposed (Li et al., 2012; O’Connor et al., 2012; O’Connor et al., 2013; Zheng et al., 2013; Foth, Tischlinger & Rauhut, 2014). While we are not able to perform new measurements, we can theorize about the effect of these newer reconstructions. Foth, Tischlinger & Rauhut (2014) confirmed Longrich’s (2006) finding of leg feathers and does not alter our results. Li et al. (2012) provide a detailed tail reconstruction of Microraptor tail plumage including a graduated tail shape with long, midline feathers; we estimate that this tail morphology may have slightly different values for stability and control effectiveness but the overall presence/absence pattern we observed here would be unchanged. O’Connor et al. (2012) and O’Connor et al. (2013) provide further information on plumage for Jeholornis; while our forewing reconstruction appears adequate, we lack the proximal fan of the “two-tailed” morphology identified in newer specimens. The additional contribution of the proximal fan is hard to estimate and should be considered in future work; however, its forward position suggests a small moment arm and therefore a small effect. This is further supported by the marginal stability and small control effectiveness of unrealistically sprawled legs, which were also near the center of mass, observed in Microraptor models (Evangelista et al., 2014b). Zheng et al. (2013) identify distal leg feathers and a long/broad tail in new specimens of Sapeornis, while Turner, Mackovicky & Norell (2012) revise the position of Sapeornis to be more basal than Jeholornis (see Fig. 4B). As tested, a more basal position for Sapeornis complicates the interpretation of our findings, thus we include a mapping onto such a tree in the .nex file at bitbucket.org/devangel77b/comparative-peerj-supplemental. However, taken together, the additional leg and tail plumage described in Zheng et al. (2013) and the more basal position proposed in Turner, Mackovicky & Norell (2012) and O’Connor et al. (2013) would maintain the patterns we see here, shifting one node earlier in the case of pitch (from node 2 to node 1).

Patterns in longitudinal stability and control

Long-tailed taxa (Fig. 5A) show a stable equilibrium point and the tail is effective in generating pitching moments, whereas short-tailed taxa (Fig. 5B) were unstable and had reduced control effectiveness of the tail. Notably, the same pattern (i.e., downward sloping Cm versus α) is seen consistently in two early representatives of the Avialae, in a long-tailed pterosaur (Rhamphorhynchus), and in the paravian Early Cretaceous dromaeosaur Microraptor, suggesting that these patterns more likely derive from shape and aerodynamics than from immediate ancestry. Similarly, the humped curves of Fig. 5B are consistent between paravians and a short-tailed pterosaur (Pteranodon).

The study taxa show progressive tail loss as well as loss of leg-associated control surfaces along with a concomitant increase in forewing size. Changes in stability and control effectiveness here (as well as manipulations in which appendage surfaces were removed with all else held constant Evangelista et al., 2014b) reflect these morphological changes. In pitch (Fig. 6), taxa shift from being statically stable ancestrally to subsequently being unstable (and thus requiring active control, or possibly damping from flapping counter-torques; Fig. 6A, red line in 6D). Control effectiveness concomitantly migrates from the ancestrally large and feathered tail (Fig. 6B, orange line in 6D) and legs to the increasingly capable forewings (Fig. 6C, yellow line in 6D), which become relatively larger, gain larger muscle attachments and gain skeletal features and stiffness proximally (Benton, 2005; Liem et al., 2000) that would improve production of left–right and fore-aft kinematic asymmetries needed for control. Distally, bone loss, joint fusion and use of ligaments to reduce degrees of freedom (Benton, 2005; Liem et al., 2000) would have enabled mechanical feedback and tuned mechanisms as flapping developed, enabling neuromuscular control effort to be focused on dealing with increasing overall flight instability and active control. Comparative forelimb myology examining extant organisms as well as a basal theropod also support the early presence of limb control via scapular protraction/retraction and pronation/supination (Burch, 2014).

Transition to forewing control co-occurs with a significantly enlarged humeral deltopectoral crest (Zhou & Li, 2010) and occurs amid progressive acquisition of a fully “avian” shoulder morphology (Turner, Mackovicky & Norell, 2012). In addition, the sternum is changing from ossified in Microraptor  through varying degrees of loss (Anchiornis and potentially Archaeopteryx) or ossification without fusion (around nodes 1–2), to ossification, with later fusion (node 3) and development of a carinate keel (node 4) (Zheng et al., in press). Concomitantly, the tail becomes much reduced into a pygostyle (Fig. 6, node 2) with increased mechanical stiffness (Pittman et al., 2013), which, combined would have decreased the moments the tail could produce and eliminated inertial mechanisms. Other synapomorphies appear at node 4 (Fig. 6), including a strut-like coracoid and triosseal canal (Benton, 2005, p 216). Whereas the latter features (node 4) feature in power production, the timing of the former features (nodes 1–2) appears more consistent with enhanced forewing control effectiveness. Ontogenetic tests (Evangelista, 2013; Evangelista et al., 2014a) show 4-day post hatching Chukar Partridge (Alectoris chukar) are capable of extreme maneuvers (rolling and pitching 180°) before strong development of a carinate sternum and before symmetric wingstrokes for WAIR, suggesting this interpretation is correct.

Roll and yaw control at high angle of attack is present early in the lineage

In roll (Fig. 7), taxa were stable at high angles of attack (Fig. 7B), but either unstable or marginally stable at low angles of attack (Fig. 7A). Large asymmetric wing movements (i.e., wing tucking) were always effective in creating substantial rolling moments early, well before development of a power stroke (Fig. 7C). Also, as animals developed the ability to fly at lower angles of attack, active control of roll would have become necessary, perhaps via inertial modes of the tail (only available before node 2) or legs, and later augmented with the forewings as they became larger and more capable of left–right asymmetry during full force production (carinate sternum, node 4). In all axes, the presence of control effectiveness early is contrary to assertions that aerodynamic functions in early and potentially aerial paravians are unimportant (Foth, Tischlinger & Rauhut, 2014). Wing movements with high control effectiveness change during evolution in a manner consistent with predicted shoulder joint mobility (criterion 1 of Gatesy & Baier, 2005). The high control effectiveness of asymmetric wing motions in roll and large inertia of ancestrally long tails echo ontogenetic changes in righting ability observed in young birds (Evangelista et al., 2014a). In maneuvering baby birds, asymmetric wing use occurs before symmetric wing use and precedes wing-assisted incline running (WAIR); the shift echoes the later evolution of symmetric wing protraction control effectiveness, although extant avian ontogenies are not necessarily linked to phylogenetic patterns in maneuvering and control.

In yaw (Fig. 8), most taxa at high angles of attack (Fig. 8A, green line in 8C) were marginally stable as might be expected from symmetry when falling near vertical. Taxa with long tails were also stable at low angle of attack (Fig. 8D, dark blue line in 8G), in agreement with computational predictions for similar shapes (Sachs, 2007). As tails are reduced, yaw stability becomes marginal and control migrates from the tail (Fig. 8E, violet line in 8G) to the wings (Fig. 8F, gray line in 8G). This is similar to what was observed in the pitch axis. Asymmetric wing pronation and supination (Figs. 8B and 8F) was effective in generating yawing moments in all taxa and at high and low angle of attack, suggesting maneuverability in yaw was present ancestrally. As the tail becomes shorter, the yaw axis becomes marginally stable or unstable (node 3), and control effectiveness must migrate (as in pitch control) from the shortening tail to the enlarging forewings. In yaw as in roll, it is possible that a carinate sternum (node 4) enables more capable left–right asymmetry during full force production in extant birds. Increased stiffness of the shortening tail (Pittman et al., 2013) would still have permitted high force production at occasional, critical moments of high angle of attack flight (Fig. 8B), such as during landing.

Stability in roll and yaw change between high and low angles of attack (Figs. 7A and 7B for roll, Figs. 8A and 8D for yaw). At high angle of attack, roll is more stable while at low angle of attack, yaw is initially stable but becomes marginal. As discussed, the control effectiveness of tail and wings also change; asymmetric wing movements are generally effective while the tail’s ability to generate aerodynamic and inertial yawing or rolling moments becomes reduced as tails are lost. The stability and control effectiveness patterns we observed illustrate a control system that works at steep angles in ancestral taxa (it is stable, but with large control effectiveness and tail inertia available), shifting to one optimized for low angles in derived taxa including extant birds (marginal, but with large control effectiveness in wings). The presence of such shifts in all axes is consistent with a transition from high glide angles to lower glide angles, as predicted by an aerial maneuvering hypothesis (Dudley & Yanoviak, 2011); it is inconsistent with a fundamental wing stroke with fixed orientation to gravity, in which angle should not matter.

Maneuvering and the evolution of flight

The findings suggest that the capacity for maneuvering characterized the early stages of flight evolution (Dudley & Yanoviak, 2011), before forewings with a power stroke fully evolved. Although early paravians may have been limited to tight coupling of vertebral and retricial movement (Gatesy & Dial, 1996), overall gross movement of the large tail of early paravians yielded high aerodynamic control effectiveness and the body possessed some degree of stability. Combined with likely dynamic forces and torques generated by either tail whipping at the root (Jusufi et al., 2008; Jusufi et al., 2011; Pittman et al., 2013) or mild asymmetric (Evangelista et al., 2014a) or symmetric forewing flapping (flapping limited by less robust skeletal and feather morphology or porosity), this suggests that early avialans and their ancestors were still capable of controlled aerial behaviors at high angles of attack (Figs. 6–8). Gradual evolution of improved maneuvering ability (increased control effectiveness, reduced stability) via consistent aerodynamic mechanisms is consistent with a continuum of aerial behaviors ranging to full aerial (Dudley & Yanoviak, 2011; criteria 3 and 4 of Gatesy & Baier, 2005). The staggered acquisition of certain morphological characters (e.g., sternum ossification; pygostyle) is consistent with aerial maneuvering evolving in an incremental and continuous manner. Subsequent shifts in control would be consistent with more shallow glides facilitated by incipient wing flapping, which may have served initially in control but then ultimately became the power stroke characteristic of modern birds. Incipient flapping may thus have become elaborated as a control response (Smith, 1952) to instabilities (Fig. 6, node 1; Fig. 8A, node 3; Fig. 8B, node 2) demonstrated here. Body center of mass was migrating forward (Allen et al., 2013), but this is coupled with loss of large posterior surfaces (long tail and leg flight feathers) and coincidence of the wing center with the COM. Active control was thus required as static stability was reduced and eventually lost (Smith, 1952), and associated forewing movements would also have enhanced aerodynamic force production and provided a means for inertial attitude adjustment. Once the transition to wing-mediated maneuverability and control began, larger surfaces and increased musculature would have facilitated dynamic force production for weight offset via the power stroke characteristic of modern birds.

Additional support for maneuvering hypotheses comes from examination of phylogenetically distant taxa. Similar trends may be present in pterosaurs (in pitch, see Fig. 5, Supplemental Information 1; bitbucket.org/devangel77b/comparative-peerj-supplemental), although we only sampled three pterosaurs here. We also tested two bats, but both appeared similar to short-tailed and extant birds (Supplemental Information 1; bitbucket.org/devangel77b/comparative-peerj-supplemental). In planform, Onychonycteris appears superficially similar to extant bats but with a slightly longer tail, and may not be a sufficiently transitional form to observe the same changes seen here. We would predict that an as-yet-undiscovered bat ancestor should possess means of effecting maneuvers such as limb inertias, enlarged tails, patagia with measurable control effectiveness, and some degree of stability at higher angles of attack; we would also predict that baby bats perform aerial maneuvers such as righting via asymmetric wing uses (which were generally effective in roll at high angle of attack), possibly in a manner similar to baby birds (Evangelista et al., 2014a). Beyond birds, bats, and pterosaurs (vertebrate fliers in a very narrow sense), the presence of highly capable aerial control systems in all the other vertebrates that “just glide”, as well as in volant invertebrates such as ants, bristletails, and stick insects, provides further support (Dudley & Yanoviak, 2011; Munk, 2011; Zeng, 2013).

Conclusions

Past hypotheses driven by cursorial scenarios and WAIR focus on symmetric forewing flapping as a means to produce forces for traction or weight support, with maneuvering and the ability to redirect such forces only occurring later. These were not supported by the patterns we observe here, which suggest a range of maneuvering ability was present from the beginning, with changes that reflect increasing aerial ability. In studies of extant animals, it is clear that, despite preconceptions some associate with terms like parachuting, gliding, and “true powered” flight, all flight includes highly dynamic situations which require means to accomplish maneuvers, whether or not force generation exceeds weight. All flight is maneuvering flight, and the patterns here are consistent with maneuvering and control playing a major role in flight evolution.

Supplemental Information

Supplemental Information 1 Supplemental figure and tables

Click here for additional data file.

Supplemental Information 2 Supplemental .nex file

Click here for additional data file.

Supplemental Information 3 Supplemental .STL files

Click here for additional data file.

Supplemental Information 4 Additional photos of pterosaur models

Click here for additional data file.

Supplemental Information 5 Additional photos of bat models

Click here for additional data file.

We thank Y Munk, Y Zeng, E Kim, M Wolf, N Sapir, V Ortega, S Werning, K Peterson, J McGuire and R Fearing for their advice and assistance. We thank the Berkeley Undergraduate Research Apprentice Program (URAP) and G Cardona, C Chun, M Cohen, E Guenther-Gleason, V Howard, S Jaini, F Linn, C Lopez, A Lowenstein, D Manohara, D Marks, N Ray, A Tisbe, F Wong, O Yu and R Zhu. The manuscript was improved using comments from ten anonymous reviewers, D Hone and N Longrich. We also thank T Libby and the Berkeley Center for Integrative Biomechanics in Education and Research (CIBER) for use of a force sensor and 3D printer.

Additional Information and Declarations

Competing Interests

Author Contributions

Data Deposition

The authors declare there are no competing interests.

Dennis Evangelista conceived and designed the experiments, performed the experiments, analyzed the data, contributed reagents/materials/analysis tools, wrote the paper, prepared figures and/or tables, reviewed drafts of the paper.

Sharlene Cam, Austin Kwong, Homayun Mehrabani and Kyle Tse performed the experiments, wrote the paper, reviewed drafts of the paper.

Tony Huynh performed the experiments, analyzed the data, wrote the paper, prepared figures and/or tables, reviewed drafts of the paper.

Robert Dudley contributed reagents/materials/analysis tools, wrote the paper, reviewed drafts of the paper.

The following information was supplied regarding the deposition of related data:

.stl files of the models and .nex files of the trees are provided as a revision-controlled Bitbucket repository at bitbucket.org/devangel77b/comparative-peerj-supplemental.

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
