# Peer review of "Shifts in stability and control effectiveness during evolution of Paraves support aerial maneuvering hypotheses for flight origins"

_PeerJ, doi:10.7717/peerj.632_

## Round 0.1 · original submission · Major Revisions

Three reviewers have given thorough and constructive reviews, and there is prospect for accepting this paper because all three of them find interesting aspects to the study. I am rendering a decision of Major Revisions because collectively their comments will require some substantial changes to the paper, which will improve its clarity and usefulness to a broad readership. Making the study as explicit and reproducible as possible will increase its impact and repute.

We will send this out for a final check by reviewers. Ensure that you include a point-by-point Response and *avoid* entering into a debate with the reviewers in the Response except where absolutely necessary. Take as many of their suggestions on board into the paper as revisions as possible. This will make their job easier on 2nd review and will satisfy the point of peer review, to improve the paper. I think the vast majority of their recommendations are very reasonable. If you want guidance from me on handling the revisions I would be happy to give advice - [email protected]

This should be a very nice addition to the literature- thanks for submitting to PeerJ!

·

Basic reporting

I am poorly placed to judge a good part of the work with regards to the wind-tunnel analysis and interpretations as my aeronautical knowledge is not that extensive, so the majority of my commentary comes with respect to the introduction and discussion, though there are some serious issues with the methods in terms of the model construction (see below).

Generally the manuscript seems sound, though there are a number of areas where I think things could be made much more clear. In several places a lack of detail is very frustrating and there are some big lists of references where it is not clear what they refer to, and use of terms that need to be much more specific or clarified in context.

Certainly a bit more in the introduction (perhaps between the current 1st and 2nd paragraphs) on control surfaces in birds / paravians would help set the context of the paper rather better. Similarly, more on the morphological transition of major features such as the shoulder and wrist and tail and their distribution among paravians would be welcome. This does not need to be too long and involved, but currently this is lacking and I think would help a lot when it comes to some of the detailed features that crop up in the discussion that previously have been unmentioned.

Experimental design

This is the area for which I have greatest concern. I think currently I would be utterly unable to render the models as used here based on the available information and thus the paper is (currently) essentially not reproducible. Given the space afforded in PeerJ generally and the presence of supplementary materials, this is really too limited right now.

Note that I am here (and below) very much sympathetic to the fact that this work was done some years ago, and it is impractical if not impossible to update the models and redo the analyses based on more recent information, but the description of the procedure and how certain decisions were justified is currently not present and should be documented as far as possible.

Currently the model building part cannot be replicated in any decent way. It is not at all clear what you did, what measurements were taken, how volumes were estimated, angles of joints reconstructed, layering and position of feathers assessed etc. The models are barely illustrated (in just one panel of a large composite figure, and seen only in dorsal view) and in a way that makes it impossible to see what they really look like and how they have been posed, and there's no information on the bats and pterosaur models at all. We seen an image of several specimens in the supplementary files and then a sort of shaded-in Archaeopteryx specimen in fig 1, and then the finished model and that's about it.

I would like to see a proper details of the construction of the models. These need not run to pages but something like a list of which specimens were used, which bits of data came from which papers, which areas were uncertain or had to be estimated etc. and especially illustrating each of the models in a couple of views will go an enormously long way to making it much more clear exactly how these were made, and will allow others to compare them to the literature / specimens.

P3, L12 citations on construction methods: Three of these are PhD theses and thus difficult to access and read, and the other doesn't contain much on the model construction so this doesn't help. I think people would struggle to use these as a source of information to repeat the methods, so getting some details into the more accessible literature would be very useful.

P3, L15-16. You imply here that you have used artistic reconstructions or skeletal reconstructions of specimens provided they were part of published papers. I would be very wary of these: they are not always accurate or appropriate even when included in peer-reviewed literature. Also, this list of papers isn't helpful they should be separated out to make it clear which papers were used for which species. For best accuracy I'd also state which bits of information can from which paper / specimen for which model (tedious yes, but if say the legs come from one paper and the tail another, this is important to know).

P3, L18. Benton, 2005 at least is not an anatomy text (it's a much more general book) and there's little here that would be of use to reconstruct a 3D model of any of these animals. There are on top of this, huge numbers of papers and book on these animals, and while I wouldn't expect you to consult unending papers on details of avian and paravian anatomy, for example the lack of inclusion of Wellnhofer's 2009 book on Archaeopteryx is odd.

P3, L21-24. Again, do this per species not as some huge list. Also, the implication here is that you used a single specimen per model is not a great approach given the vagaries of preservation. Not a lot you can do about it now, but again, at least make it clear.

P3, L32-34. This would be impossible to trace for a reader: say which paper falls into which category.

Fig 1d. It is almost impossible to tell at this scale but some of the proportions look rather out here on the models (hence why it is important to document what you have done and show better figures of these models). Anchiornis for example seems to have metatarsals less than half the length of the femur when they should be c. 80-90% of femoral length. Why has Archaeopteryx got separate primaries and not Microraptor or others? Why are there no feathers on the proximal arm of Microraptor but these are included on Archaeopteryx (neither to my knowledge has this part of the arm well preserved in terms of feathers)? These are some major questions and potentially big issues with the reconstructions, but I simply can’t assess them because the information is not currently available either in terms of the models themselves or how certain decisions were made in their reconstruction.

Validity of the findings

As noted above, I do have some concerns about the models that may impinge on the results, but this is hard to say with the current information. Some of the use of the methods and how they are presented are odd as well. Specific comments on this follow:

P5, L15. This is an odd approach. For such a simple tree with so few taxa in it, why was it assembled from 4 separate phylogenies, and how was it done? Which papers contributed which parts of the tree and why? Or does the same effective tree appear in every paper, in which case this should be rewritten to reflect that.

P5, L22-25. What are these characters and where is the data? I’ve not been able to identify them. I’m struggling to follow this generally– you say there are “eight discretized stability values (stable, marginal, unstable, coded according to if the 95% confidence interval of measurements includes zero or not)”. What are these 8 characters? What were the divisions for the discreet states and how were these selected? Also why have you ‘discretized’ some characters? Mesquite can handle continuous data, so surely you are just removing signal prior to the analysis.

Additional comments

In addition to the above comments, below are various specific comments and suggestions for the manuscript. Much of this is about clarification of issues and feed into many of the issues raised above.

Abstract: I’m not sure why there are so many references in the abstract: I’m not used to seeing any.

Abstract: “Results of aerodynamic testing are mapped phylogenetically to examine how maneuvering characteristics correlate with tail shortening, fore- and hindwing elaboration, and other morphological features.”
I’m not sure you actually do this. While you discuss these features, this does imply some specific correlations were found and if so these need to be spelled out more clearly. In particular, at times in the discussion you discuss changes to the anatomy at various nodes, so why not make this more obvious and explicit with a figure that shows which morphological transitions occurred at which notes and this will make them much easier to compare to your figures 3-6.

P1, L5. What are these species? Give an example.

P1, L4-8. This sentence with these clauses is not very clear.

P1, L15. There are numerous specimens of Microraptor exhibiting this condition, as this paper makes clear. Do you mean a single *species*? Even here the new Changyuraptor, but also things like Anchiornis, Pedopenna and even Archaeopteryx, suggest this is not uncommon among paravians.

P1, L17-18. Make this more explicit, which taxa changed to what degree?

P2, L11-13. It’s not clear what these papers are referring to.

P2, L17. “was developing” when, in what context?

P2, L26. Hanging close parenthesis.

P2, L28. This is the first figure mentioned in the paper, so it should be figure 1, not figure 4. Renumber accordingly.

P2, L35-36. What would this look like if it were discovered? That gives a frame of reference for understanding the difference.

P2, L36. How is the wrist joint a "rigidly fused/fixed surfaces located posteriorly"?

P2, L37. “these are not observed” You haven't said what they are yet, only that one of them is a semi-lunate carpal. In any case, that doesn't meant they are not present in other, related taxa. It will be much more convincing if you can demonstrate these features are not in any of these lineages.

P2, L46. Presumably these are studies in birds, rather than control generally.

P4, L18. I would split this into two paragraphs here.

P5, L6-8. This is a major point, supply a reference to support it.

P5, L32-35. This simply repeats what is in the methods section and is redundant.

P6, L2. “slightly different” Higher or lower? And why?

Discussion: Obviously the timing is just unfortunate that this came out after your paper was submitted, but you do need to take a look at the Changyuraptor paper and what it says about tail function.

P6, L18-23. Do you mean the whole tail as a unit with feathers, or just potentially based on the mass of it acting as a lever? The patterns seen for Rhamphorhynchus in Fig 3a at least suggest it may have nothing to do with feathers given how different the soft tissue construction of the tail is from this to paravians (proportionally small and distal flap that is D-V orientated). What about the comparisons to other species mentioned earlier? Various bats and pterosaurs are surely ideal for comparison here since you've done the work already.

P6, L32. I find it hard to believe this is the best / most recent source on these kinds of transitions and you must include page numbers when citing a whole book. I had to skim several chapters to find this and check it.

P6, from L37 I'd be careful of over interpretation here - you do have relatively little data. I don’t think there is anything wrong with the interpretations based on what you have collected, but assuming these things co-occur at certain nodes when you are looking at just a handful of taxa is obviously going to depend very much on which species you have sampled (there is nothing from the huge group of Enantiornithes for example) and how typical they are, and I suspect there may be a lot more subtelty to this, and this should at least be recognised. This is especially true when the prime source of character acquisition you seem to be relying on is essentially a review from nearly 10 years ago. Surely there are much more recent and better phylogenies and lists of character changes you can refer to here?

P6, L50. ‘early’. You need to clarify things like this – do you mean early in time, or early in the phylogenetic lineage? Archaeopteryx likely had some form of power stroke, but the ones we are really confident about are from the Early Cretaceous, but things like Anchiornis date from the Middle Jurassic. That’s only one or two nodes in most phylogenies, but between 5 and 20 million years.

P7, L3. ‘Channels’ What do you mean here? This is the only time this word is used in the MS and I really don't know what it is supposed to refer to in this context.

P7, L4. Really? I doubt control did a lot for many species, or do you mean *flying* paravians? There’s plenty of deinonychosaurs that probably were not gliding.

P7, L8. Symmetry of what?

P7, L12. But if this is also present in Microraptor and Anchiornis, then it occurred prior to avian origins, not early in the avian lineage surely?

P7, L36. ‘Ancestral’ needs clarification: ancestral to what? Avialae, Paraves? From the context you seem to mean ancestral to paravians, implying that these might be volant in some capacity.

P7, L42. ‘quantum’ is an odd word here, ‘continuous’ is better.

Figure 1. Why the small panels? Make each of these a figure (and as above, greatly increase the coverage of the models). The text is too small and almost impossible to read as well.

Figures 4-6. These are too small and moreover really hard to read, the colours are all too similar and the overlapping lines are really hard to spot and read easily. I’d separate these all out into a multi-panel figs with just one colour / character per tree.

In these figures I'm not sure why you have labeled Microraptor and Anchiornis as 'outgroups'. The term doesn't appear elsewhere in the MS and would normally be used to designate sister-taxa that are outside the realm of consideration and there to provide a putative ancestral condition to the ingroup.

·

Basic reporting

See below

Experimental design

No comments

Validity of the findings

Findings are justified by the experiments

Additional comments

This is an interesting and important paper on an understudied issue in the evolution of flight- the evolution of stability. There are a few places, especially the figures and discussions, where things could be improved a bit but I would encourage publication and think this is likely to become an important and cited paper; I would be happy to recommend publication with minor revisions.
It is sometimes claimed that the flight stroke is “the central problem” in avian evolution but this is misleading; there are a whole host of problems an animal has to solve- stability and control, lift generation, weight reduction, thrust generation, and increased metabolic output- to fly effectively, and there’s the issue of the sequence in which these adaptations are assembled. In that sense there are a series of key innovations that are needed to create something that can fly effectively, not one key innovation. This paper deals with one aspect of the problem- stability- that hasn’t received nearly enough attention.
The approach used here- scale models used in a wind tunnel with force gauges- is appropriate and while the models are unlikely to perfectly mimic real animals, that isn’t really the point; they allow us to simplify the problem and figure out how one variable- here, airfoil geometry- affects flight ability in terms of stability.
The results are perhaps not surprising; it’s been argued before that stability preceded instability, and given the large control surfaces in basal forms and the small ones in more derived forms, the results are entirely what you’d expect. Still, it’s important as the only study, as far as I know, to really quantify this and look at it in a phylogenetic context.
One weakness of the paper are the figures, the phylogenies are a little confusing because they are attempting to map out several things on one small tree. If space is not a consideration, then I would suggest trying to do several different trees, one for each character being mapped, and then put the character being mapped next to each taxon (e.g. “Microraptor- marginal stability). As it stands, I find it difficult to read the figures and figure out what is happening.
I think the paper would also benefit a little from discussing what these results mean in terms of the evolution of flight. In general, hypotheses of flight evolution fit into two main camps- an arboreal or trees-down scenario, and a cursorial, ground-up scenario (WAIR would be a variant of this). Yes, this will undoubtedly annoy certain people, but working on flight evolution tends to do that.
Arboreal evolution scenarios (of the sort advocated by Darwin, Norberg, Feduccia, etc.) hypothesize that flight evolution followed a pattern similar to what we see in gliding animals- arboreality, followed by parachuting, followed by gliding, with the ability to flap to produce thrust occurring last, and only in a handful of lineages. Cursorial scenarios (including WAIR) envision flapping occurring early; wings are initially used produce thrust, and only later do lift generation and control become more important.
The different scenarios make very different predictions about the sequence of character acquisition. If the arboreal scenario is correct, then control appears early, in fact it is one of the first things to evolve. The large bushy tail and skydiver posture in parachuting red squirrels, for example, appears before the gliding patagium seen in flying squirrels, suggesting that squirrels first develop the ability to control their descents and later the ability to extend leaps with lift production.
I would argue that the authors’ results fit nicely with an arboreal scenario. Control, especially at high angles of attack, seems to appear early, later we get the ability to produce thrust. This fits in very nicely with the scenario outlined by Norberg. I would recommend a brief paragraph summarizing and referencing some of the different scenarios and laying out how this does or does not fit the predictions of these hypotheses.
It would be interesting to spend a paragraph or two discussing the broader context. These results may ultimately turn out to be applicable to other groups, e.g. pterosaurs and insects. Last, it would be worth discussing why instability is so important. The authors mention improved maneuverability as one possibility, but it seems to me that many short-tailed birds (e.g. anseriformes) aren’t terribly maneuverable, and many highly maneuverable birds (e.g. swallows, hawks) have fairly large tails. Flying wings might provide an informative analogy- they are unstable but not terribly maneuverable, instead eliminating vertical and horizontal stabilizers reduces weight and drag, improving fuel efficiency and increasing range over conventional aircraft.

Reviewer 3 ·

Basic reporting

Results section is very short - please elaborate / summarize trends in graphs

No Conclusions section?

A few run-on sentences but otherwise fine

Experimental design

Generally fine. Bats are measured but never discussed?

Validity of the findings

Statistically, can you provide some metric of confidence in your ancestral states / node reconstructions? How confident are you in your character coding? I noticed on your supplemental tables that for several of your taxa, the standard deviations are equal in magnitude or greater than some of your measurements (especially roll, low alpha). Please comment.

Additional comments

Pg. 1 (of text), line 20: “ asymmetric and symmetric flight feathers are absent from the legs”. Careful! I know what you mean here but maybe state differently - raptors, etc. do have contour feathers on the legs.

Pg. 2, line 17: “ While the capacity to generate aerodynamic forces for weight support (Dial, 2003; Dial et al., 2008; Heers et al., 2014) was almost certainly developing,”. Perhaps replace “developing” with “evolving”?

Pg. 2, line 23: “ Morphologically, for an organism able to perform aerial control with only a weak power
stroke, we would expect some surface, though not necessarily large, and perhaps with limited force
25 generation due to reduced speed, or feather porosity and flexibility”. This sentence is a little confusing.

Pg. 2, line 28: figure 4 is referenced before any other figures

Pg. 2, line 30: “(Dial, 2003; Dial et al., 2008; Heers et al., 2014)”. These papers do not state that stability and control were absent during the early evolution of flight.

Pg. 2, line 31-35. I’m not sure why having large muscles precludes stability and control? Please clarify.

Pg. 2, line 39: “ it is possible for some appendage motions or positions to ineffective or unstable, as in the case of certain sprawled postures and leg movements in Microraptor (Evangelista et al., 2014, though these were also anatomically infeasible).” This sentence is a little confusing.

Pg. 3, line 17 - should probably include genus and species of Starlings.

Pg. 3, line 21. How did you account for mass? Presumably your 3D printed models were more light-weight than a real animal? They may produce the same amount of aerodynamic force, but the wing-loading would be reduced… How does this affect your interpretation? Aw, I see you mention this later on.

Also, how did you estimate muscle mass to get 3D body outlines? This will have a big effect on your COM estimates - please explain / justify.

Pg. 3, line 24: what about the effects of feather micromorphology (transmissivity, stiffness, etc) that would not be accounted for by using paper wings? Since you’re looking at trends it’s probably fine, but perhaps point that out.

Pg. 4, line 48 (pitching moment equation). I’ve never personally measured pitching moments, but I’m a little unclear on why lamba is snout-vent length, rather than distance to COM? Or there is no “distance” to COM because you’re measuring all planforms at once, rather than considering tail, wings etc. separately? Perhaps clarify for people unfamiliar with this equation?

Pg. 5, lines 8-13: These predictions are very important - perhaps include them in the introduction? Would be useful to read early on.

Pg. 5, line 23. 8 stability values and 12 control? I see “stable, marginal, and unstable”, but it’s not clear here where the 8 comes from?

Pg. 5, line 30. Your Results section is very short. Please expand by summarizing graphs - what trends you notice? etc.

Pg. 6, line 24: “ The study taxa show progressive tail loss as well as loss of leg-associated control surfaces along with a concomitant increase in forewing size”. Perhaps begin the section with this sentence, to remind readers of phylogenetic trends in morphology?

Pg. 6, line 40, etc. You mention nodes several times, which I can see in the figures - but you do not mention which morphological features are changing at which nodes - please clarify.

Pg. 7, line 21 (and line 42): “ The presence of such stability shifts is more consistent with a transition from high glide angles to lower glide angles”. Which stability shifts are you talking about? I’m a little confused... Are you saying that being more stable in roll than in yaw is consistent with high glide angles? Why? Please clarify.

Figure 3. What do the different colored dots mean? Easy to miss the text - add a legend?

Figure 4, etc. - can’t read figure legends.

In the figure 4 legend, you state that control effectiveness of the wings (protraction / retraction) is reduced in more derived taxa, but then you state that control has shifted from the tail to the wings. Don’t these two statements contradict each other?

---

## Round 0.2 · Minor Revisions

Good news, the 3 reviewers agree that the paper is almost acceptable and with minor revisions we can accept it!

Hone makes the very good point that any additional information you can provide on the pterosaur models' designs will be of broad value and enhance the paper for the community's benefit. If you can make any improvements there, please do.

Longrich gives some helpful pointers on improvements to the figures in particular, which I agree with. Figure 1's white spaces should be reduced by rescaling the images, and Fig 4 could have a stacked on top of b, allowing each to fill the pace to maximize visibility of the animal images. FIg 6 needs slightly better labelling to make it as foolproof clear as you can; Figs 7's + 8's formats should be matched to that, too.

Reviewer 3 objects to “Alternatively, both stability and control may have been absent early in the evolution of flight, only appearing after a strong and bilaterally symmetric power stroke evolved ” using citations of two Dial papers as support (or straw man, as they contend). I see no value in citing these here, either, so it's best to remove them. It does not harm the paper to do so and might avoid colleagues taking umbrage at unfair treatment. Similarly, a subsequent statement citing (Dial, 2003; Dial et al., 2008) seems to misrepresent that work or set up a false dichotomy between control and power strokes. Other clarifications are provided in the reviewer's pdfs and these seem justifiable as tips for revision.

Please try to make all of these changes- they seem manageable and fair. But if you wish to discuss them with me via email or rebut them in the Rebuttal document, that is OK too.

I look forward to receiving the revised MS and getting this published!

·

Basic reporting

This manuscript has improved considerably since the original version I reviewed and I am pleased that the authors have attended to many of the details and suggestions that I and other referees originally proposed. In particular, the methods section has been improved with respect to the models and I do feel this is important. Although I agree with the authors and referee Nick Longrich that it is perhaps impossible to reconstruct such animals to a very high degree of accuracy (and that this may not matter too much to general results), this is important. First of all such details allow any major differences between reconstructions to be easily identified and it also allows for (theoretically at least) the models to be accurately replicated by other groups for additional testing – if both models and results of an analysis are very different, it is very hard to determine which is generating different data, hence the extra information is important, even if (as suggested) minor details are likely relatively unimportant to overall flight profiles.

On that note, there is still no information at all on the pterosaur models and these are not even illustrated in the supplementary files – given that we do not have extant descendants of pterosaurs to work from and the major disagreements (and indeed advances) in issues such as the wing structure, wing chord, wing posture etc. of pterosaurs (e.g. see Elgin et al.,2011; Palmer, 2010) again not having even a single image to work from does make it very hard to judge their validity. While I imagine they are still useful, if nothing else it seems a shame that you have generated these models, and then done the analysis, but have little to show for it when I am sure these would be of great interest given the relatively limited work done to date on pterosaurs in wind tunnels.

Experimental design

No comments.

Validity of the findings

No comments.

Additional comments

Two minor issues:

P2, L65. The ‘predatory strike hypothesis’ is rather problematic (e.g. see Sullivan et al., 2010 on wrist evolution), and even though this is rather tangential to the main point in this line, is worth revising. Even Padian at the time (2001) noted it was not clear how this might actually work functionally.

P9, L393. Typo: ‘Avialanas’

·

Basic reporting

OK

Experimental design

OK

Validity of the findings

OK

Additional comments

Overall, the paper looks good and I recommend publication. Personally I think it would be worth spending more time discussing what this means for cursorial/WAIR versus arboreal parachuting/gliding hypotheses, but that's of course up to the authors how they want to spin this.

In terms of figures, I still have a few suggestions but these are more along the lines of minor tweaks prior to publication, rather than something that would require a revise-and-resubmit. I think Figure 1 could be tightened up a bit... I think it would look better if either adjacent panels in each row were scaled to the same height, or else the panels in each column were scaled to the same width; a multi-panel figure should look like a comic book page with only narrow gutters of whitespace between adjacent panels.

Figure 4 is hard to read- the text is small and the pictures of each bird are reduced almost to invisibility. It would be good to either enlarge Fig. 4 and/or trace the birds in Adobe Illustrator to make little silhouettes for each taxon...

Figure 6 is still a little hard to read. I would consider putting a short vertical bar across the stem where you think each functional shift is changing. That's sort of the established visual shorthand for saying "this feature evolved here on the tree".

Reviewer 3 ·

Basic reporting

See attached file

Experimental design

See attached file

Validity of the findings

See attached file

Additional comments

See attached file

Annotated reviews are not available for download in order to protect the identity of reviewers who chose to remain anonymous.

---

## Round 0.3 · accepted · Accept

I'm accepting this, as the changes are definitely fine- except that I don't see the changes to the figures- did you upload the old ones, perhaps? e.g. Fig 1 doesn't look like it fills the page as described, and Figs 6-8 lack "tick-marks" to note where traits map on the phylogenies-- to my eyes, the figures look ~the same but it's late at night... so please double-check and ensure PeerJ gets the correct figures (email the main PeerJ address if you have trouble re-uploading anything now that I'm accepting the paper). I am accepting the paper pending those figure checks and finalizing.